# Multi-View Visual Question Answering with Active Viewpoint Selection

**DOI:** 10.3390/s20082281

**Published:** 2020-04-17

**Authors:** Yue Qiu, Yutaka Satoh, Ryota Suzuki, Kenji Iwata, Hirokatsu Kataoka

**Affiliations:** 1Graduate School of Science and Technology, University of Tsukuba, Tsukuba 305-8577, Japan; yu.satou@aist.go.jp; 2National Institute of Advanced Industrial Science and Technology (AIST), Tsukuba 305-8560, Japan; ryota.suzuki@aist.go.jp (R.S.); kenji.iwata@aist.go.jp (K.I.); hirokatsu.kataoka@aist.go.jp (H.K.)

**Keywords:** visual question answering, three-dimensional (3D) vision, reinforcement learning, deep learning, human–robot interaction

## Abstract

This paper proposes a framework that allows the observation of a scene iteratively to answer a given question about the scene. Conventional visual question answering (VQA) methods are designed to answer given questions based on single-view images. However, in real-world applications, such as human–robot interaction (HRI), in which camera angles and occluded scenes must be considered, answering questions based on single-view images might be difficult. Since HRI applications make it possible to observe a scene from multiple viewpoints, it is reasonable to discuss the VQA task in multi-view settings. In addition, because it is usually challenging to observe a scene from arbitrary viewpoints, we designed a framework that allows the observation of a scene actively until the necessary scene information to answer a given question is obtained. The proposed framework achieves comparable performance to a state-of-the-art method in question answering and simultaneously decreases the number of required observation viewpoints by a significant margin. Additionally, we found our framework plausibly learned to choose better viewpoints for answering questions, lowering the required number of camera movements. Moreover, we built a multi-view VQA dataset based on real images. The proposed framework shows high accuracy (94.01%) for the unseen real image dataset.

## 1. Introduction

Recent developments in deep neural networks have resulted in significant technological advancements and have broadened the applicability of human–robot interaction (HRI). Vision and language tasks, such as visual question answering (VQA) [1,2,3,4,5,6], and visual dialog [7,8], can be extremely useful in HRI applications. In a VQA system, the input is an image along with a text question that the system needs to answer by recognizing the image, interpreting the natural language in the question, and determining the relationships between them. VQA tasks play an essential role in various real-world applications. For example, in HRI applications, VQA can be used to connect robot perceptions with human operators through a question-answering process. In video surveillance systems, the question–answer process can serve as an interface to help avoid the manual checking of each video frame, significantly reducing labor costs.

Although VQA methods can be useful in real-world applications, there are numerous problems related to their implementation in real-world environments. For example, conventional VQA methods answer a given question based on a single image. However, in real-world environments, because it is challenging to take photographs continuously from optimal viewpoints, objects can be greatly occluded and thus answering questions based on single-view images could be difficult. Considering that multi-view observations are possible in HRI applications, this study discusses VQA under multi-view settings. Qiu et al. [9] proposed a multi-view VQA framework that uses perimeter viewpoint observation for answering questions. However, perimeter viewpoints may be difficult to set up in real-world environments due to environmental constraints, making their method difficult to implement. In addition, observing each scene from perimeter viewpoints is relatively inefficient, especially for applications that require real-time processing. Moreover, the authors did not evalutate their method under real-world image settings.

Here, we propose a framework, shown in Figure 1, in which a scene is actively observed. Answers to questions are obtained based on previous observations of the scene. The overall framework consists of three modules, namely a scene representation network (SRN) that integrates multi-view images into a compact scene representation, a viewpoint selection network that selects the next observation viewpoint (or ends the observation) based on the input question and the previously observed scene, and a VQA network that predicts the answer based on the observed scene and question. We built a computer graphics (CG) multi-view VQA dataset with 12 viewpoints. For this dataset, the proposed framework achieved accuracy comparable to that of a state-of-the-art method [9] (97.11% (Ours) vs. 97.37%) that answers questions based on the perimeter observation of scenes from fixed directions, while using an average of just 2.98 viewpoints as a contrast with 12 viewpoints of the previous method. In addition, we found that our framework learned to choose viewpoints efficiently for answering questions by ending observation while the observed scene contains all objects needed to answer the question, or additionally observing the scene from viewpoints with large spacing to enhance accessible scene information, thereby lowering the camera movement cost. Furthermore, to evaluate the effectiveness of the proposed method in realistic settings, we also created a real image dataset. In experiments conducted with this dataset, the proposed framework outperformed the existing method by a significant margin (+11.39%).

The contributions of our work are three-fold:we discuss the VQA task under a multi-view and interactive setting, which is more representative of a real-world environment compared to traditional single-view VQA settings. We also built a dataset for this purpose.we propose a framework that actively observes scenes and answers questions based on previously observed scene information. Our framework achieves high accuracy for question answering with high efficiency in terms of the number of observation viewpoints. In addition, our framework can be applied in applications that have restricted observation viewpoints.we conduct experiments on a multi-view VQA dataset that consists of real images. This dataset can be used to evaluate the generalization ability of VQA methods. The proposed framework shows high performance for this dataset, which indicates the suitability of our framework for realistic settings.

## 2. Related Work

### 2.1. Visual Question Answering

In recent years, a variety of VQA methods have been proposed. Holistic VQA methods, such as feature-wise linear modulation (FiLM) [10], combine image features with question features by concatenation [1], bilinear pooling [6], or the Hadamard product [3], and then predict answers from these combined multi-modal features.

A series of VQA datasets have been proposed. The VQA 2.0 dataset [2], one of the most popular datasets, contains real images sampled from the MS COCO dataset [11], and question–answer pairs collected using crowdsourcing (Amazon Mechanical Turk).Johnson et al. [12] pointed out that the VQA2.0 dataset has high dataset distribution biases and proposed the Compositional Language and Elementary Visual Reasoning (CLEVR) [12] dataset, in which images and questions are generated by pre-defined programs. The CLEVR dataset has been widely adopted for evaluating visual reasoning ability. Although significant progress related to VQA tasks has been made, there is relatively little discussion about multi-view VQA. Qiu et al. [9] proposed a multi-view VQA framework. However, the authors used a limited perimeter viewpoint setting, and their study did not conduct real-world environment experiments. Here, we propose a framework with high efficiency in terms of the number of observation viewpoints and discuss its application in real-world image settings.

### 2.2. Learned Scene Representation

Traditional approaches based on three-dimensional (3D) convolutional neural networks (CNNs) learn a 3D representation from 3D data, such as point cloud data [13,14], mesh [15], and voxel data [16]. These methods can extract representations that contain the underlying 3D structure of the input data. However, due to the discrete nature of 3D data formats, such as point clouds, the resolution of the learned representation is limited. In addition, learning from direct 3D data usually requires a massive amount of training data, high memory cost, and a long execution time.

In contrast, a series of continuous 3D scene representations have recently been proposed. Generative query network [17] is a network based on a conditional variational autoencoder [18] that learns a meaningful 3D representation described by the parameters of a scene representation network (SRN) from a massive amount of 3D scene data annotated with only camera viewpoint information. Deep signed distance function [19] is an auto decoder-based structure that learns continuous signed distance functions to map 3D coordinates into the signed distance of a group of objects. Scene representation networks [20] proposed by Sitzmann et al. predict 3D range maps along with a scene representation of the input scene and thus exhibit robustness to unknown camera poses.

A learned continuous scene representation contains the underlying 3D structure and object information of the input scene and usually requires a relatively small amount of annotation signals and relatively short execution costs. Therefore, in this work, we use a continuous SRN to extract scene information and use the representation to answer questions.

### 2.3. Deep Q-Learning Networks

Q-learning [21] is a type of value-based reinforcement learning framework that holds a Q-table of the values of taking actions *a* in environment states *s*. However, for a large number of environment states *s* or continuous state spaces, the time and space consumption of Q-learning can become rather high. Instead of holding a Q-table in memory, deep q-learning networks (DQNs) [22] use neural networks to predict action values from environment states *s*. Conventional DQN methods that involve image information learn policies from raw image inputs [23]. In the present work, instead of raw images, we use scene representations that contain underlying 3D structure and object information of scenes and encoded question information as input and then adopt a DQN to select observation viewpoints from the previously observed scene and question information.

### 2.4. Embodied Question Answering

The work most similar to ours is the embodied question answering (EQA) task defined by Das et al. [24]. The authors define an EQA task for which an agent is embodied within a 3D environment. The agent attempts to answer given questions by navigating and gathering needed image information using egocentric perceptions of the environment. The authors proposed an EQA baseline method that splits the EQA task into a recurrent neural network-based hierarchical planner-controller navigation policy for navigating to the target region in the environment and a VQA model that combines the weighted sum of the image features of five images and the question features to give a final answer. Yu et al. [25] proposed a multi-target EQA task, in which every question involves two targets. In [26], the authors extended the original CG EQA dataset setting to photorealistic environments and suggested the use of point cloud data.

Most EQA methods focus on the vision-language grounded navigation process and finding the shortest paths in a given environment. They usually use relatively simple VQA methods, such as in [24], where the authors simply concatenate five images into a VQA model for answering questions. In the above study, relatively little attention was given to the kind of information necessary to answer questions, and no evidence was given that the framework answers questions based on a 3D understanding of the given scene. In contrast, our work focuses on exploring necessary scene information to answer questions by selecting observation viewpoints. Therefore, our approach can avoid unnecessary observation and reduce camera movement costs. In addition, our framework is based on an integrated 3D understanding of the given scene observed from multiple viewpoints. Therefore, our framework can associate 3D information with viewpoints and determine the minimum necessary scene information required to answer questions via the selection of observation viewpoints.

## 3. Approach

In real-world HRI applications, it can be challenging to obtain photographs from perimeter viewpoints. In addition, it is efficient to end the observation when the input scene information is sufficient to answer the question. Based on the above, we propose a framework that actively observes the environment and decides when to answer the question based on previously observed scene information.

As shown in Figure 2, the proposed framework consists of three modules, namely an SRN, a viewpoint selection network, and a VQA network.

The inputs of the overall framework are the default viewpoint $v_{0}$, the image $x_{0}$ of the scene observed from $v_{0}$, and the question *q*. $v_{0}$ and $x_{0}$ are first processed by the SRN to obtain the original scene representation $s_{0}$. The question is processed by an embedding layer and a two-layered long short-term memory (LSTM) [27] network to obtain $q^{\prime }$. Then, the viewpoint selection network predicts the next observation viewpoint (or selects the end action) based on $s_{0}$ and $q^{\prime }$. If viewpoint $v_{t}$ is chosen at time step *t*, the agent obtains the image $x_{t}$ from viewpoint $v_{t}$ of the scene (e.g., for a robot, a scene image from $v_{t}$ is taken). Next, the SRN updates the scene based on $x_{t}$ and $v_{t}$. If the end action is chosen, the VQA network predicts an answer based on the $q^{\prime }$ and the integrated scene representation at that time. In the following sections, we discuss these three networks in greater detail.

### 3.1. Scene Representation

We use the SRN proposed by Eslami et al. [17] to obtain integrated scene representations $s_{t}$ from viewpoints $\left\{v_{0},v_{1},\ldots ,v_{t}\right\}$ and images $\left\{x_{0},x_{1},\ldots ,x_{t}\right\}$. For scene *i* observed from *K* viewpoints, the observation $o_{i}$ is defined as follows:(1)$$o_{i}={\left\{\left(x_{i}^{k},v_{i}^{k}\right)\right\}}_{k=0,\ldots ,K-1}\;$$

The scene representation $s=f_{SRN}\left(o_{i}\right)$ and $g(x^{m}|v^{m},s)$ is jointly trained for image rendering from arbitrary viewpoint *m* to maximize the likelihood between the predicted $x^{m}$ and the ground truth images. $f_{SRN}$ integrates multi-view information into a compact scene representation. We use the above framework to train the SRN.

### 3.2. Viewpoint Selection

For VQA in real-world environments, it is necessary to choose an observation based on the question and previously observed visual information. For example, for the question “ is there a red ball?”, if a red ball has previously been observed, the question can be answered instantly. Additionally, for highly occluded scenes, it may be necessary to make observations from a variety of viewpoints. Therefore, we propose a DQN-based viewpoint selection network $f_{VS}$ to actively choose actions.

More specifically, assuming that the scene can be observed from *K* viewpoints, we define an action set $A={\left\{a_{i}\right\}}_{i=0,\ldots ,K}$, where $a_{0},\ldots ,a_{K-1}$ denote the viewpoint selection actions and $a_{K}$ represents the end observation action.

The viewpoint selection network $f_{VS}$ predicts a *K*+one-dimensional, vector that represents the obtainable reward value of each action (after it is executed) from the input of the previously observed scene $s_{t}$ and question $q^{\prime }$. Assuming that the *j*-th action $a_{t}$ is chosen at time step *t*, we denote the predicted reward value $r_{t}^{eval}$ of action $a_{t}$ under the environment state $s_{t}$ as follows:(2)$$r_{t}^{eval}={\left\{f_{VS}\left(s_{t},q^{\prime }\right)\right\}}_{j}\;$$

The real reward value of action $a_{t}$ can be formulated as Equation (Equation 3), where $r_{t}^{env}$ denotes the reward obtained from the environment, and γ is the discount factor of future rewards:(3)$$r_{t}^{real}=r_{t}^{env}+\gamma argmax_{a_{t+1}}f_{VS}\left(a_{t+1}|s_{t+1},q^{\prime }\right)\;$$

The overall objective of viewpoint selection is to minimize the distance between $r_{t}^{eval}$ and $r_{t}^{real}$.

We designed the reward $r^{env}$ based on the correctness of the question answering and the numbers of selected viewpoints. For each added new observation viewpoint or viewpoint that is chosen repeatedly, we assign penalties $p_{sp}$ and $p_{rt}$ (hyperparameters). We denote the VQA loss as $loss_{vqa}$ (normalized to [−1,1]). We designed $r^{env}$ for three action types, as shown below.
(4)$$r^{env}=\left\{\begin{array}{c} p_{sp}+p_{rt}-loss_{vqa}\;\left(i\right)\\ p_{sp}+p_{rt}\;\left(ii\right)\\ -loss_{vqa}\;\left(iii\right) \end{array}\right.\;$$

For the final viewpoint selection action, the reward is (i); for the other viewpoint selection actions, the reward is (ii); for the end action, the reward is (iii).

### 3.3. Visual Question Answering

VQA predicts an answer based on integrated scene information *s* and the processed question $q^{\prime }$. We denote the VQA network to be trained as $f_{VQA}$. The answer ans can be predicted by the following:(5)$$ans=argmax_{ans}f_{VQA}\left(ans|s,q^{\prime }\right)\;$$

The network is optimized by minimizing the cross-entropy loss between the predicted ans and the ground truth answer. In this study, we used the state-of-the-art FiLM method [10] as the VQA network. However, it is noteworthy that the VQA framework can be arbitrary.

The proposed framework could not deal with ill-structured or non-English questions that are not included in the training dataset. However, the framework could be expanded for these situations by further integrating sentence structure checking or translation modules.

## 4. Experiments

We conducted experiments using settings with CG and real image datasets, respectively. In the following sections, we discuss the experimental details for these two settings.

### 4.1. Implementation Details

Our overall framework is composed of three modules, namely an SRN, a viewpoint selection network, and a VQA network. Here, we introduce the implementation details of these three modules. The implementation setting is used for both CG and real image experiments. We used PyTorch [28] as the implementation framework. rtwoWe also show the detailed network structure in Figure 3.

**SRN:** we adopted a 12-layered tower-structured SRN proposed in [17] to extract scene representations. The SRN transforms the input of 64×64×3-dimensional images and 1×1×7-dimensional camera information (a vector represents camera position and rotation) into a 16×16×256-dimensional scene representation. We used Adam optimizer [29] and an initial learning rate of $5\times 10^{-4}$ for training. In all experiments, we trained the SRN network for 140 epochs.

**Viewpoint Selection Network:** to extract question information, we first adopted an embedding layer (torch.nn.Embedding defined in PyTorch), which transforms the one-hot vector of each word into a 300-dimensional vector. Next, we adopted a two-layered LSTM with hidden dimensions of 1024 to encode each question sentence into a 1024-dimensional vector. Alternative structures, such as the gated recurrent unit [30] or transformer [31], can also be used for question feature extraction. Next, we combined the encoded question with the scene representation by concatenation, which resulted in a 1024+16×16×256-dimensional feature. Next, we processed the concatenated feature via three fully connected layers to predict a *K*+one-dimensional, action reward value vector (K indicates the viewpoint number). We adopted a tangent activation function before the final fully connected layer. We set the hyperparameter γ of the viewpoint selection network to 0.8 and ϵ-greedy to choose actions. The initial ϵ was set to 0.5 and multiplied by 1.04 every three epochs. The penalties for viewpoint action and repeated action were set to –0.25 and –0.05, respectively. This network was jointly trained with the VQA network.

**VQA:** we modified the FiLM implementation code [32] for our VQA module. We removed the image feature extraction component used in [32] and used integrated scene representation as the input image feature. The question extraction component is shared with the viewpoint selection network during training. We jointly trained the modified FiLM network with the viewpoint selection network with an initial learning rate of $3\times 10^{-4}$ and Adam optimizer for 40 epochs in all experiments.

### 4.2. Experiments with CG Images

**Dataset Setting:** we modified the CLEVR dataset generation program by placing multiple virtual cameras in each scene and then generating a multi-view CLEVR dataset (Multi-view-CLEVR_12views_CG). The camera setup, which is widely used in multi-view object recognition tasks (e.g., [33,34]), is shown in Figure 4 (left). The cameras are elevated 30 degrees above the scene and placed at intervals of 30 degrees around the center of the scene. We place a relatively large object in the middle of the scene and three to five other objects around the center object. The setting was designed to make it difficult to observe all objects from a single viewpoint, thereby providing a simple testbed for viewpoint selection efficiency. We created CG objects and set the illumination, lighting, background of scenes, and virtual camera positions through a 3D creation software blender [35] and Python-blender source code based on the the CLEVR dataset generation program [36]. We set two types of questions: exist (query the existence of an object) and query color (query an object’s color). Then, by considering the existence of relationship words (left of, right of, behind, in front of), the questions can be further separated into spatial and non-spatial types. Example images and questions of a scene from Multi-view-CLEVR_12views_CG dataset are shown in Figure 4 (right) and the detailed statistics of the dataset are shown in Table 1.

**Quantitative Results:** the overall and per-question accuracies are shown in Table 2. Our framework (SRN_FiLM_VS) showed a slight drop in overall accuracy compared with that for SRN_FiLM proposed in [9]. However, our framework used many fewer viewpoints on average (2.98 vs. 12). This indicates that the proposed framework learned to obtain the required information for answering questions, lowering the camera movement cost. This ability, which is especially crucial for applications with restricted observation viewpoints, makes our framework more efficient because it decreases the required number of camera (or robot) movements.

**Qualitative Results:** the results for four example questions are shown in Figure 5. For Question 1, the queried object “tiny cube” appears in the default view, so our model ended viewpoint selection instantly and answered the question. For Questions 2 and 3, the queried objects do not appear in the default view, so our model selected an additional viewpoint for observation. These results show that our model learned how to make observations to obtain the necessary scene information for answering questions. For Question 4, our model used three additional viewpoints and gave an incorrect answer. In this case, the queried object was a purple cylinder. The shadow cast by the center object made it difficult for the model to recognize the color. It is believed that using higher-resolution input images could improve performance in such situations.

**Effect of Viewpoint Selection:** we conducted an experiment using the input of randomly sampled viewpoints (SRN_FiLM_Random) and equally sampled viewpoints (SRN_FiLM_Equal), in which all the viewpoints from Multi-view-CLEVR_12views_CG were evenly sampled. The results are shown in Table 3. We first evaluated the accuracy of answering questions based on a randomly picked viewpoint (SRN_FiLM_Random (1 view)). For this setup, an overall accuracy of 66.71% was achieved, which indicates that our dataset is challenging with single-view images. Moreover, because our viewpoint selection network predicts answers based on an average of 2.98 viewpoints, we sampled three viewpoints for equity. The results show that our viewpoint selection network greatly outperforms the randomly and equally sampled settings, indicating that the proposed viewpoint selection network makes the utilization of viewpoint information more effective through constructing scene representations needed for answering questions from the partial observation of scenes.

**Runtime information:** the test process was conducted on an Intel(R) Core(TM) i9-7920X CPU @ 2.90 GHz machine with a graphic processing unit of Nvidia TITAN V. The runtime is 0.043 s per scene on average with a single GPU and single process.

### 4.3. Experiments with Real Images

#### 4.3.1. Training on CG Images and Testing on the Real Images Dataset

Real-world environment applications, such as HRI applications, require the ability to sense and recognize real-world scenes. Thus, the ability to transfer the learned scene representation and understanding obtained in a simulated environment to a real-world scene is important for AI-based frameworks. To evaluate the transfer ability of our framework, we built a multi-view dataset with real images of objects similar to those in the CLEVR dataset. We then evaluated the performance of the model that learned in the simulated environment (multi-view CLEVR dataset) on the real image dataset.

**Dataset Setting:** to evaluate the model performance for real images, we built a dataset (Multi-CLEVR_4views_Real) that consisted of photographed real images. We printed out and colored models with attributes similar to those of the objects described in the previous section using a 3D printer. We then placed these models on a scene table of a laboratory with normal fluorescent lighting. We used a camera (Sony Alpha 7 III Mirrorless Single Lens Digital Camera [37]) to take photographs of scenes from four viewpoints. The camera was elevated 30 degrees above the table and placed at 90-degree intervals around it with a distance of 1.1 m from the scene center.

An average of seven photographs were taken for each viewpoint, with the camera slightly moved and rotated. In total, we collected 100 scene photographs, which were augmented to obtain 500 scene photographs by sampling images taken randomly from each viewpoint. The question–answer pair settings were the same as that those used in the previous section. The statistics are shown in Table 1. Several scene examples are shown in Figure 6 (middle).

**Quantitative Results:** we first trained and tested the SRN_FiLM and our framework on Multi-view-CLEVR_4views_CG (Figure 6 (left)) for 40 epochs (the statistics are shown in Table 1). The test split results are shown in Table 4. Our framework achieved performance comparable to that of SRN_FiLM while using fewer viewpoints. We then evaluated the performance for the Multi-view-CLEVR_4views_Real, the results of which are shown in Table 5. Compared with the test results for CG images (Table 4), those for the real images dropped by nearly 30% for both methods without fine-tuning (first and fifth rows of Table 5).

#### 4.3.2. Fine-Tuning on the Semi-CG Dataset

This reduction in performance is likely caused by the domain gap between CG and real images, such as color distribution, lighting, and object texture differences. To obtain high performance for real images with models that were directly trained using simulated images, a common approach is to fine-tune the models with real images. However, in our data setting, each scene is composed of objects with random attributes and locations and is observed from four viewpoints. Therefore, photographing a real scene under our dataset setting requires manually arranging objects and moving cameras, which is both labor- and time-consuming.

To solve the problem mentioned above, we introduce a method for creating a fine-tuning dataset that scans real objects and imports the obtained models into a simulation environment. In detail, the dataset construction process includes three steps. First, we placed the real objects used for building the Multi-view-CLEVR_4views_Real in a 3D photography studio (3D MFP, Ortery [38]) for scanning. The 3D MPF photographs object models from multiple viewpoints. Next, we used the software 3DSOM Pro V5 [39] to generate textured 3D models, which is accomplished by aligning and integrating multiview images obtained by the 3D MFP. Finally, we imported the obtained 3D object models into the CLEVR image generation process and created a dataset, which we denoted as Multi-view-CLEVR_4views_3DMFP, using the same scene and camera setup used for Multi-view-CLEVR_4views_Real. We use the Multi-view-CLEVR_4views_3DMFP dataset for fine-tuning (the statistics of which are shown in Table 1, example images are shown in Figure 6 (right)).

Next, we conducted ablation experiments on the fine-tuned network parts (freezing the VQA model, freezing the SRN model, and then fine-tuning both models). The results are shown in Table 5. We found that fine-tuning the models improved their performance by a large margin, especially when both the SRN and VQA models were fine-tuned (up to +27.19%). After fine-tuning, the proposed model achieved an accuracy of 94.01% on the unseen real image dataset, outperforming SRN_FiLM by 11.39% (fourth and eighth rows of Table 5). This performance increase is attributed to the proposed SRN_FiLM_VS model being relatively less affected by noise in images from unnecessary viewpoints, which enables the model to adapt faster to unfamiliar scenes. Furthering discussion is left for future work. It is noteworthy that using the fine-tuning dataset Multi-view-CLEVR_4views_3DMFP helps significantly improve the performance of both models. This result shows that, by introducing the “semi-CG” dataset consisting of the scanned real object models and simulation environments for the fine-tuning process, real-world applications have the potential to improve the performance of models trained from pure simulation data.

**Qualitative Results:** Here, we discuss the qualitative results shown in Figure 7. The SRN_FiLM method answered questions based on all four viewpoints; in contrast, our framework selected the observation viewpoints (highlighted in red bounding boxes). For Questions 1, 2, and 3, the proposed model made reasonable viewpoint selections and answered the questions correctly. For Question 4, the spatial reasoning was difficult, and both models failed to give the correct answer. Narrowing the gap between the CG and real images might improve performance.

## 5. Conclusions

In this study, we proposed a multi-view VQA framework that actively chooses observation viewpoints to answer questions. VQA is defined as the task of answering a text question based on an image that is essential in various HRI systems. Existing VQA methods answer questions based on single-view images. However, in real-world applications, single-view image information can be insufficient for answering questions, and observation viewpoints are usually limited. Moreover, the ability to observe scenes efficiently from optimal viewpoints is crucial for real-world applications with time restrictions. To resolve these issues, we propose a framework that makes iterative observations under a multi-view VQA setting. The proposed framework iteratively selects additional observation viewpoints and updates scene information until the scene information is sufficient to answer questions. the proposed framework achieves performance comparable to that of a state-of-the-art method on a VQA dataset with CG images and, in the meantime, greatly reduces the number of observation viewpoints (a reduction of 12 to 2.98). in addition, the proposed method outperforms the existing method by a significant margin (+11.39% on overall accuracy) on a dataset with real images, which is closer to the real-world setting, making our method more promising to be applied to real-world applications. However, directly applying a model trained on CG images to real images results in a performance gap (a drop of −30.82% on accuracy). In the future, we will consider using various methods to narrow this gap.

## Figures and Tables

**Figure 1 sensors-20-02281-f001:**
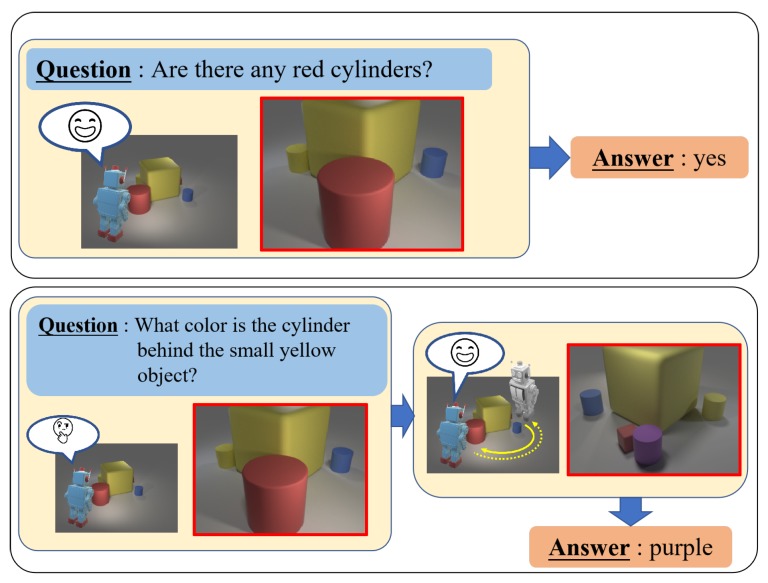
Illustration of multi-view VQA with active viewpoint selection. Our framework selects the observation viewpoint (highlighted in red bounding boxes) to answer questions based on the observed scene information.

**Figure 2 sensors-20-02281-f002:**
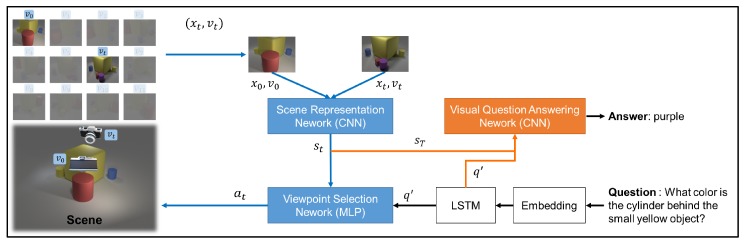
Overall framework. The viewpoint selection network (multi-layer perceptron (MLP)-structured) chooses the observation viewpoint (or ends the observation) iteratively based on the scene $s_{t}$ and question $q^{\prime }$ (blue arrow flow). When the end action is chosen, the scene representation up to that time step $s_{T}$ and the encoded question $q^{\prime }$ will be processed by the VQA network to provide an answer (orange arrow flow).

**Figure 3 sensors-20-02281-f003:**
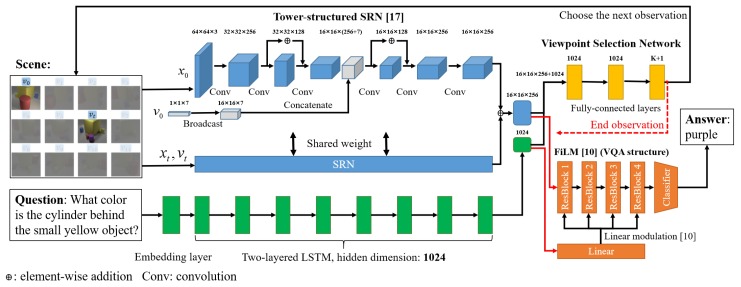
Detailed network structure. Our overall framework is composed of a tower-structured SRN [17] (shown in blue), a viewpoint selection network (shown in yellow), and a VQA framework (shown in orange), which is modified from FiLM [10]. The viewpoint selection network and VQA share the question feature extraction structure (shown in green). Based on the previous observation, the viewpoint selection network chooses the next observation or ends observation (red dotted arrow flow). While the end observation is chosen, the scene representation to that time step, and the question feature (red arrow flows) will be processed by the VQA framework to give an answer prediction:

**Figure 4 sensors-20-02281-f004:**
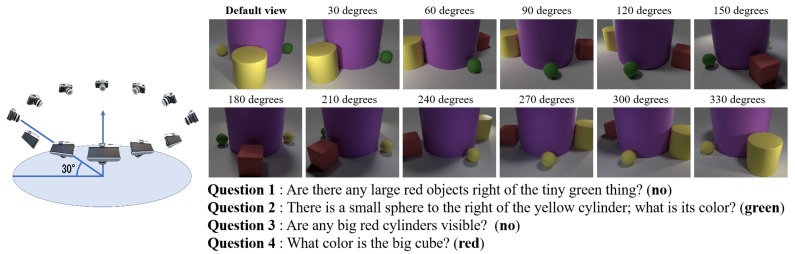
Virtual camera setup (left) and example images and questions from Multi-view-CLEVR_12views_**CG** dataset (right). The default view determines the spatial relationships of objects in each scene.

**Figure 5 sensors-20-02281-f005:**
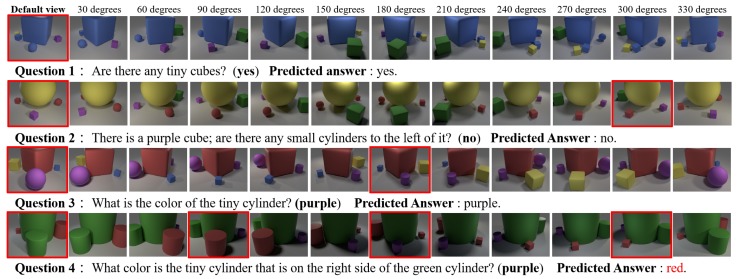
Example results of SRN_FiLM_VS for the Multi-view-CLEVR_12views_**CG** dataset. The default view and selected view images are highlighted in red bounding boxes. The incorrect answers are shown in red.

**Figure 6 sensors-20-02281-f006:**
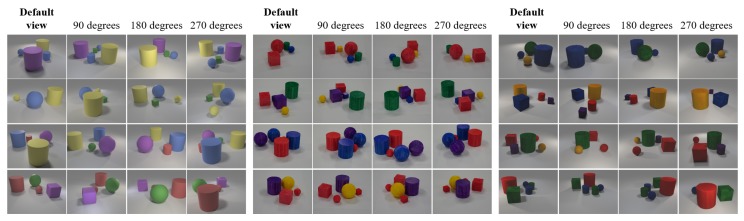
Example images from Multi-view-CLEVR_4views_**CG** (**left**) which is entirely built from simulated scenes, Multi-view-CLEVR_4views_**Real** (**middle**), which are created from photographing real table scenes composed of real object models, and a Multi-view-CLEVR_4views_**3DMFP** (**right**) dataset that is built from importing scanned real object models into simulated scenes and aimed for fine-tuning.

**Figure 7 sensors-20-02281-f007:**
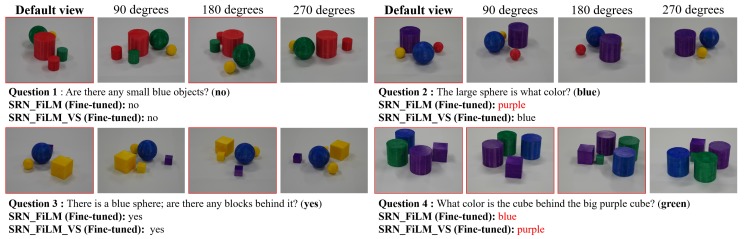
Example results for Multi-view-CLEVR_4views_**Real** dataset. The default view and selected view images are highlighted in red bounding boxes. Incorrect answers are shown in red.

**Table 1 sensors-20-02281-t001:** Statistics for datasets used in this study.

Datasets	No. of Scenes	No. of Q–A Pairs	Spatial		Non-Spatial	
	(Train/Test)	(Train/Test)	Exist	Query Color	Exist	Query Color
Multi-view-CLEVR_12views_CG (**CG**)	(40,000 / 3,000)	(157,232 / 11,801)	43,000	42,993	43,000	40,040
Multi-view-CLEVR_4views_CG (**CG**)	(40,000 / 3000)	(158,860 / 11,928)	43,000	42,997	43,000	41,791
Multi-view-CLEVR_4views_3DMFP (**CG**)	(10,000 / 0)	(39,694 / 0)	10,000	9994	10,000	9,700
Multi-view-CLEVR_4views_Real (**Real**)	(0 / 500)	(0 / 1974)	500	500	500	474

**Table 2 sensors-20-02281-t002:** Evaluation results for Multi-view-CLEVR_12views_**CG**.

Method	Overall Accuracy	Spatial		Non-Spatial		Average no. of Used Viewpoints
		Exist	Query Color	Exist	Query Color	
SRN_FiLM	**97.37%**	94.20%	98.20%	99.03%	98.11%	12
SRN_FiLM_VS	97.11%	95.27%	96.90%	99.03%	97.25%	2.98

**Table 3 sensors-20-02281-t003:** Effect of viewpoint selection on accuracy.

Method (No. of Used Viewpoints)	Overall Accuracy
SRN_FiLM_Random (1 view)	66.71%
SRN_FiLM_Random (3 views)	81.39%
SRN_FiLM_Equal (3 views)	82.90%
SRN_FiLM_VS (2.98 views)	97.11%

**Table 4 sensors-20-02281-t004:** Evaluation results for Multi-view-CLEVR_4views_**CG**.

Method	Accuracy	No. of Used Viewpoints
SRN_FiLM	97.67%	4
SRN_FiLM_VS	97.64%	**2.02**

**Table 5 sensors-20-02281-t005:** Evaluation results for Multi-view-CLEVR_4views_**Real**.

Method	Fine-Tuning		Accuracy
	SRN	FiLM	
SRN_FiLM	-	-	67.88%
SRN_FiLM	-	✓	76.14%
SRN_FiLM	✓	-	77.30%
SRN_FiLM	✓	✓	82.62%
SRN_FiLM_VS	-	-	66.82%
SRN_FiLM_VS	-	✓	79.99%
SRN_FiLM_VS	✓	-	91.56%
SRN_FiLM_VS	✓	✓	94.01%

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
