# Peer review of "Multi-View Visual Question Answering with Active Viewpoint Selection"

_sensors, 2020, doi:10.3390/s20082281_

Round 1
Reviewer 1 Report
Interesting project and good work but I would suggest the following to address lack of detail on methodology and specific measures used -
In line 47+ there is mention of performance similar to a state of the art system. The state of the art system needs to be described. Additionally, you mention in line 49 that your system learned to use better viewpoints. This should be explained.
Line 206 - describe what the gap is between CG and real images.
Line 305 - what is the significant margin? Statistical analysis? Line 307 - how substantial is the performance gap?
It would be helpful to more clearly describe the environment and tools used - camera, graphics, software, time/duration of observations, etc. This would strengthen your methodology. Also - more clearly explain why your approach is promising. The conclusion didn't provide a strong summary of your paper. Realize many will read the conclusion to determine if they want to read your paper. Give them a reason to read your paper. Your abstract is good - write a conclusion that is strong like your abstract.
Author Response
First of all, we thank the reviewers for the constructive comments. We did our best to revise the manuscript, including the parts that they pointed out. In our revised manuscript, we highlight the parts modified regarding comments from reviewer 1 in blue, and the parts modified regarding the comments from reviewer 2 in orange. Please see the attachment for the details of our responses to the comments.

Reviewer 2 Report
In this work, the authors are proposing a new multi-view question answering framework that allows the observation of a scene from multiple angles to answer specific questions. Their proposition seem novel to me and is compared with other states of the art methods with very promising numerical results on benchmark datasets. Furthermore, the article is well written and the main references from the state of the art are covered, although the dedicated section could be longer and give more details on the main differences between these systems.
My major questions and remarks are the following:
1) The proposed architecture describes one of his component as being "LSTM-based" (Figure 2 and legend). What does LSTM-based mean ? Is it LSTM ? is it not ? A sub-question would be : Why using LSTM in this context ? Is it just to remember past questions (in which case something should loop back to the LSTM network in figure 2). And if so, the question is open on whether LSTM is a better solution than just remembering past answers. Could there be alternative "one shot" architectures that take one question at the time ? Would it still work with the rest of the proposed architecture ?
2) All neural networks component used should be detailed: how many layers, what size, what activations functions and what optimizer ?
3) Related to remark 2, unless the authors are using a pre-made architecture with only minor modifications (it should be made clear in the text), or just pluging together existing blocks of a pre-made architecture, the choice of the components and neural network parameters from remark 2 should be justified.
4) Two naïve questions:
- How to decide where is the right and where is the left when you have 360 views ? Does the "default view" determine it ? How is it handled ? Does the network "know" the angle and use it to decide ?
- how does the system react if you ask it a question that is ill structured, makes no sense, or is not even english ?
Minor comments:
1) Figure 3 and 4 are really small. If possible, the author might want to consider changing the setting.
Author Response

(The authors gave the same response as above.)

Round 2
Reviewer 2 Report
I am satisfied with the modifications proposed by the authors and I therefore recommand this paper for publication.
Figure 3 and 4 are much better.
One minor element that could still be improved : You added some elements to describe the network on page 7 and a reference to the paper from which the original architecture comes from. However, I still believe that the full structure of the network should be explicitely given in the paper, and not just a reference. In this regard, a figure detailing the different layers might be more efficient than a text paragraph.
Best of luck for the continuation of this work !
Author Response
We appreciate the reviewer's comments and recommendations. We added a new figure (Figure 3. in the modified manuscript) that describes the detailed network structure of our framework. Please confirm the Figure 3. The added caption of Figure 3 is shown below: > Detailed network structure. Our overall framework is composed of a tower-structured SRN[17] (shown in blue), a viewpoint selection network (shown in yellow), and a VQA framework (shown in orange), which is modified from FiLM[10]. The viewpoint selection network and VQA share the question feature extraction structure (shown in green). Based on the previous observation, the viewpoint selection network chooses the next observation or ends observation (red dotted arrow flow). While the end observation is chosen, the scene representation to that time step, and the question feature (red arrow flows) will be processed by the VQA framework to give an answer prediction. We also added a brief description of Figure 3. in line 196 as below (the added description is shown in green): > We also show the detailed network structure in Fig. 3. In addition, the figure numbers of the original Figure 3-6 are modified to 4-7. We thank the reviewers again for their careful reading of the manuscript and their constructive comments, recommendations.